# Indicators as Mediators for Environmental Decision Making: The Case Study of Alessandria

**Alessandra Battisti *** , **Maria Valese and Herbert Natta**

Department of Planning, Design and Technology of Architecture, University of Rome—La Sapienza, 00185 Rome, Italy; maria.valese@uniroma1.it (M.V.); herbert.natta@gmail.com (H.N.)
* Correspondence: alessandra.battisti@uniroma1.it

**Abstract:** The design of urban public open spaces plays a key role in the development of micro-scale reactions to global phenomena (pandemic, climate change, etc.) that are currently reshaping the human habitat. Their transformability and healthy influence on the urban environment make them strategic nodes for acupunctural regeneration with systemic effects. Several methods, models, and indicators have been developed to face the complexity of these spaces, made up of tangible and intangible layers; however, there is a gap between theoretical investigation and the need for public administrations to devise feasible solutions, strategies, and guidelines. The paper focuses on this mediation, presenting, as a case study, an adopted methodology and the first results achieved according to guidelines for the regeneration of the system of squares in the historical center of Alessandria (Piedmont, Italy). In this case, a multidisciplinary approach and a Multi-Criteria Analysis (MCA) method, supported by geospatial analysis and GIS technology, have been employed to work as mediators for a participatory process which will involve public administration, stakeholders, experts, and researchers. The paper presents an overview of the workflow, with a focus on the first set of thematic indicators and an open conclusion. It will explain how they have been defined, integrated, and turned into a dialogic tool, with the aim of laying the foundation for the next stage of involvement by the public administration and stakeholders. Specific attention will be paid to the key role of vegetational and environmental parameters, which represents the requalification strategy's backbone, for both local and systemic scales.

**Keywords:** open spaces; multi-criteria analysis; participation; decision making; GIS



## 1. Introduction

The United Nations (UN) Sustainable Development Goals (SDG) includes the enhancement of inclusiveness, safety, resiliency, and sustainability of cities and human settlements (SDG-11), with a specific target placed on the accessibility of green and public spaces (Target 11.7) [1]. To monitor actions, threats, and achievements related to this goal, UN-Habitat has developed a Global Public Space Program (GPSP), which stands on an inclusive definition of public space as «all places publicly owned or of public use, accessible and enjoyable by all for free and without a profit motive» (*Charter of public space*, Art. I.6) [2]. Starting from this general statement, the GPSP guidelines recognize that the streets are the typology of space which most fulfils these criteria [3] (p. 27), defining the "character of a city", and generating a "connective matrix" that «forms the skeleton of the city upon which all else rests" [1] (p. 3).

Streets are defined as the kind of public spaces that guarantee the essential urban function of mobility by breaking the physical limit of buildings. In this broad sense, this category includes avenues and boulevards along with squares and plazas [3] (p. 27). In addition to their connective function, the streets play a fundamental social role, being aggregation places both in everyday life and for extraordinary events. Consequently, their aesthetic/environmental quality has a strong impact on the general *healthiness* of

people [4–6]. Furthermore, «they can be defined as multi-use public spaces» [3] (p. 27) with high transformative potential.

In 2020, in a note related to the COVID-19 pandemic, the UN-Habitat stressed the key role of public spaces, «whether to limit the spread of the virus or to provide ways for people to relax or carry out their livelihood» [7], emphasizing their multi-functionality and adaptability. The design of public spaces is also a core part of «sustainable urban development» [8] (p. 24). Adapting to the changing climatic conditions links the maintenance of the usability of public spaces to the optimization of their bioclimatic performances. For example, the growing temperatures increase the phenomenon of Urban Heat Islands (UHI), with documented consequences on human health [9–11]. Thus, public spaces' management, maintenance, and design have to deal with their complex nature, involving the built, social, and natural environment, and considering tangible and intangible layers, physical components, temporary uses, and symbolic and identarian meanings. Consequently, it requires adequate multidisciplinary and systemic approaches and methods to integrate contributions from different fields of study, providing decision-makers with holistic and synthetic development scenarios.

This need for mediating between decision making and the complex nature of the urban environment dates back to the foundation of urban planning as a discipline; however, ways of realizing this link have changed due to technological evolution and epistemological paradigm shifts.

In the first half of the 20th Century (except for some pioneering earlier studies), the language of mathematics started to be more systematically used for describing the city. At the intersection between economy, geography, and social sciences, in the framework of location theory, several models have been developed to analyze the relationship between economic activities and their spatial position, providing strategies to select locations for maximizing profits [12] (p. 4).

This quantitative approach, along with the positivist idea of the urban space as a measurable object, has been progressively criticized and revised in the second half of the 20th century, in a qualitative revolution grounded in the Structuralist cultural context. The city appeared in its complexity [13], leaving behind the Renaissance and Modernist illusion of an ordered system.

The interconnection between urban space and social life was certainly not new [14–18], but at the core of this Copernican revolution was a new centrality paid to the active role of people in urban planning (through participatory practices) and livability as a priority objective for urban design [19–21]. This perspective did not exclude mathematical modelling, but it pushed the definitions and integration of multidisciplinary systems of indicators (SI), paving the way for a parametrization trend that has progressed until recently.

The main goal of this line of investigation is to measure the current or potential value [22] of a place, considering not only its "internal impact" [23] (p. 5), determined by economic parameters, but also the correlation between value and place quality, including indicators related to health, environment, society, and culture [24].

Several livability SIs have been developed [25,26], combining parameters from different domains (such as natural, security, education, economy, health, leisure, culture, transportation, etc.) [27], to measure this correlation.

For urban scales, this correlation brought a more general interpretation to 'quality of life', in that it should be inclusive of human factors related to social equity environmental sustainability [28], to measure the «perceived residential urban quality» [29].

However, the applicability of these SIs to urban planning, and the replicability of these theoretical models in different contexts, when moving from a global scale to a local one, suffered limitations related to (1) the availability of the required data for elaborating upon the indicators; (2) the cultural shift, that hardly permits an unambiguous interpretation of indicators; (3) the complexity of the SIs, and the consequent difficulties in integrating with urban planning and decision-making workflows [30,31].

In this sense, the widespread diffusion of digital technologies, both in everyday life and in professional activities, with no exception for urban planning, opened the door for rethinking SIs more as participatory tools rather than top-down models.

In fact, considering the city as the Relational Environment of a Third Infoscape [32], where people constantly, consciously or unconsciously, generate data, «it is possible to gain insights about how to make interventions in it» [32] (p. 6), understanding «the resulting flows of knowledge, wisdom, emotion, and, in general, communication» [32] (p. 8).

An example of this conceptual framework is the experimental project *Mercé* (300.000 km/s), where the definition of a SI becomes a participatory process that involves citizens in the training of an Artificial Intelligence (AI) algorithm to design more livable cities [33].

Furthermore, observing the characteristics and flow of digital information helps to understand the non-linear behavior of the city, where the physical environment works as a framework for spontaneous, generative, emergent, and temporary phenomena.

In this sense, the literature attests well to the use of Location-Based Social Networks (LBSN) to analyze human activity in the urban environment [34], revealing behavioral patterns [35], and providing information for developing qualitative indicators related to place, ordinary or temporary use, attractiveness, perception, and so on [36–38].

However, this growing availability of data, and their integration in urban analysis, still leaves a gap between knowledge and action [9,39], as it misses a further integration step in terms of decision-making workflows. From one side, the need for public space to have sustainable development strategies, and to be able to resist and react to environmental urban transformations, moves the local PA (Public Administration) to look for adequate design solutions; from the other side, the proposed SIs hardly match with the procedures of decision-makers [40]. In this way, the SI remains more a theoretical framework for urban analysis than an operative tool for urban planning.

This paper aims to fill the knowledge gap between knowledge and action by implementing a novel research framework through (1) analyzing socio-spatial patterns of historical urban fabrics and (2) using the analysis results to assess urban generation. At the core of our proposal is the consideration that the SI is not a prescriptive but dialogic tool; it is a base for collecting and integrating input from different actors in a participatory process that includes researchers, experts, stakeholders, and citizens. The renovation of the system of squares in the historical center of Alessandria (Piedmont, Italy), promoted by the local municipality, works as a case study to test the workflow and discuss further development.

## 2. Materials and Methods

### 2.1. The Case Study

Alessandria is a city that is inhabited by 100,000 people, located in the Northwest of Italy, at the top of the Pianura Padana geographical region, and in the middle of the so-called industrial triangle (generated by the productive centralities of Turin, Milan, and Genoa, Figure 1).

The administrative area of the municipality of Alessandria is far bigger than the urban settlement (the 8% of a 204 km$^2$ total surface): marked by the path of two rivers (Tanaro and Bormida), it presents the characteristics of an alluvial plain, with prevalent agricultural land use (80%).

The official foundation of the city dates back to the Middle Ages (1064), as a sort of synoecism of previous settlements in a fortified city, blessed by Pope Alexander III against the emperor, Frederick I. Located as a fortress on the natural boundary of the river Tanaro, the city has had two main vocations in its history: a military use, which is still perceivable in of some of its public spaces, including the compact of historical center and its main monumental attraction (the fortress called Cittadella); and a key commercial and industrial role, as it was a strategic infrastructural node (both for the railway and for the highway).

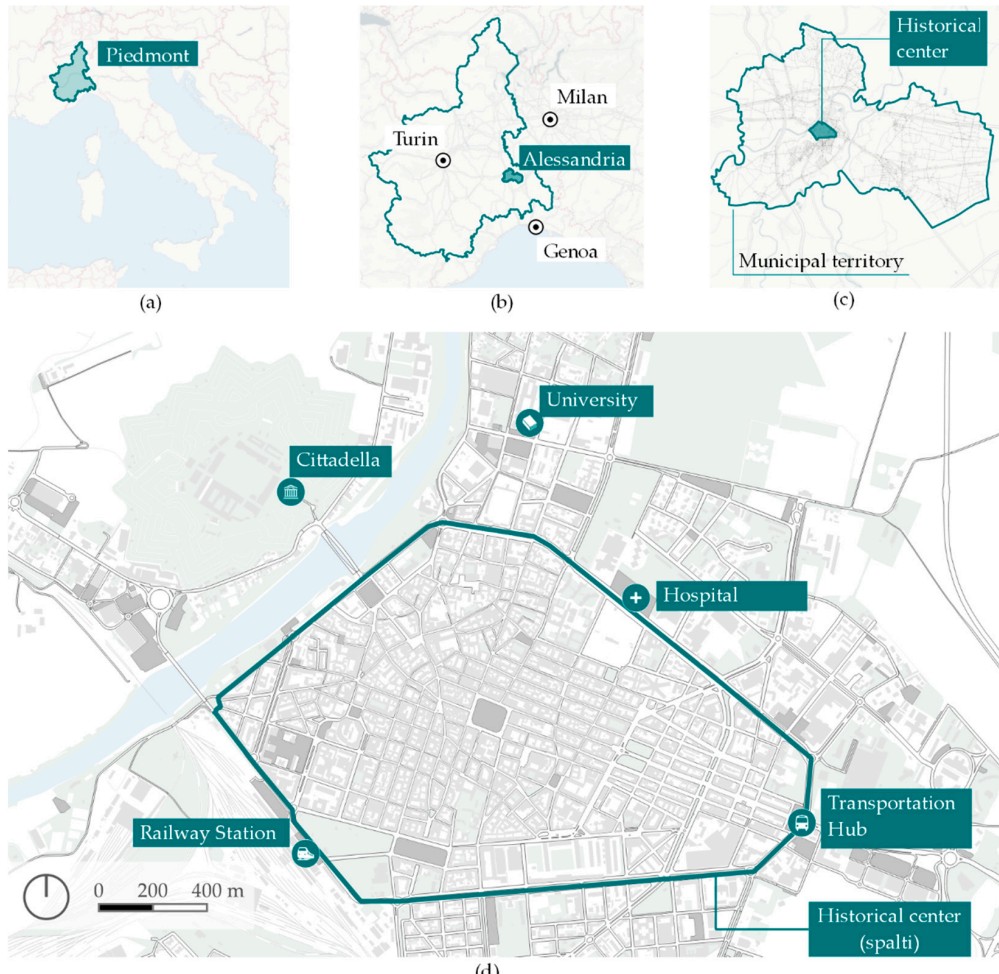

**Figure 1.** Alessandria is located (**a**) in the North-West of Italy; (**b**) in the middle of the industrial triangle generated by the major cities of Turin, Milan, and Genoa; (**c**) its historical center covers a small part of a wider municipal territory; (**d**) the majority of the green areas are out of the boundaries of the historical center, which is mostly a built environment and its open public spaces are mainly covered by parking areas; however, the size of the historical center is compatible with the concept of a 15 min walkable city [Sources: Authors, OpenStreetMap, Municipality of Alessandria].

The shape of the historical center of the city is clearly defined by these three factors of the urban morphogenesis: the natural element of the river that directly touches the settlement in the North-West, dividing it from the Cittadella, the historical trace of the fortified walls, that have been removed since the 19th century but remain in the pathway of the so-called *spalti* (urban expressways), and the railway station (in the South-West).

The urban texture of the historical center is highly densified, with a percentage of the built environment being at around 50% (0.7 km$^2$ on the 1.5 km$^2$ total surface are of the historical center) whereas only 3% are green areas. The city's biggest parks are outside of the perimeter of the *spalti*, and are located on the riverside, in front of the railway station, and to the North-East, close to the cemetery and the airport. Thus, the majority of open spaces in the city center are paved, with a prevalent use of parking areas, where the green elements appear as ornamental spots in the public areas and as small gardens in the private ones.

In 2020, the municipality of Alessandria set the renovation of the public spaces in the historical center as a priority objective, identifying 13 squares to be designed, considering both their functional role in the urban system and their specific identity as individual urban components (Figure 2).

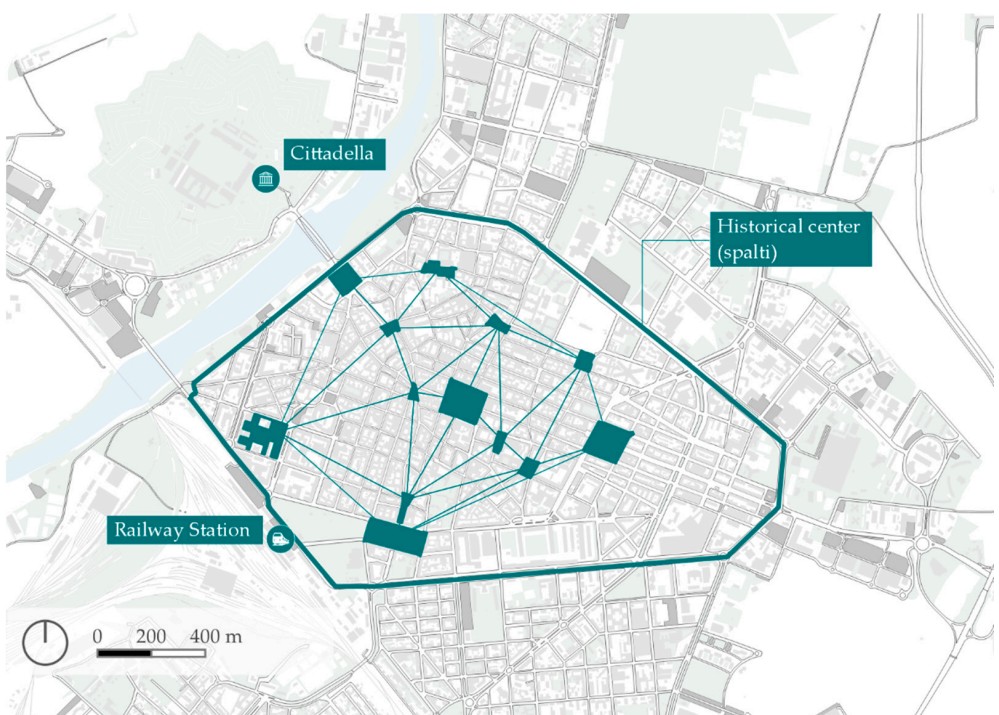

**Figure 2.** The 13 selected squares are located in both the inner part of the historical center and on its boundaries, and they include different kinds of spaces, from the oldest historical squares to places that are not even proper squares, but a system of parking areas [Sources: Authors, OpenStreetMap, Municipality of Alessandria].

*2.2. Methods*

2.2.1. Methodological Framework

The adopted methodological approach to provide development strategies for the public spaces in the historical center of Alessandria is framed, in the general idea of public spaces, as infrastructure for urban regeneration.

The squares, in particular, are public spaces characterized by (1) *connectivity*, they are nodes of a network of streets that permit movement through the urban environment; (2) *aggregation*, they are not just places to cross, but also places to stay, with fundamental socio-cultural and economic roles, hosting formal and informal community events along with services and economic activities; (3) *healthiness*, they have to be comfortable places for people, maintaining a high level of environmental and aesthetic quality.

Thus, to manage the specific complexity of the object of study, where invisible and intangible phenomena overlay and interact with the physical dimension, it has been necessary to adopt a multidisciplinary perspective, and consequently, a methodological framework to convert heterogeneous analytic outcomes to a synthetic vision of a development scenario.

Considering this main goal, the research team proposed a workflow based on the Multi-Criteria Decision Analysis (MCDA) methodological framework: a branch of Decision Analysis (DA), developed to find an optimal solution in conditions where there are multiple conflicting objectives [41–43].

The applications of MCDA ranged from economy to management, with no exception for urban planning [44], having both a descriptive and a prescriptive function in modelling and optimizing the decision-making process.

The above-defined specificity of public spaces requires the model to be adapted [45,46] and taken in its more general and flexible interpretation; however, the main advantage of its use is the possibility of integrating heterogeneous contributions from different stakeholders in a formalized and transparent workflow, from the analytic to the decisional phase.

The process started with an initial domain definition, provided by the Municipality, in an official call for regeneration guidelines, which include: seven topics (accessibility, mobility, education, society and culture, landscape and environment, tourism, commerce), correspondence with its internal operative organization, and the expected outcomes and objectives.

Starting from this preliminary input, we developed a workflow structured in four phases: (1) mapping, (2) evaluation, (3) elaboration, and (4) scenario (Figure 3).

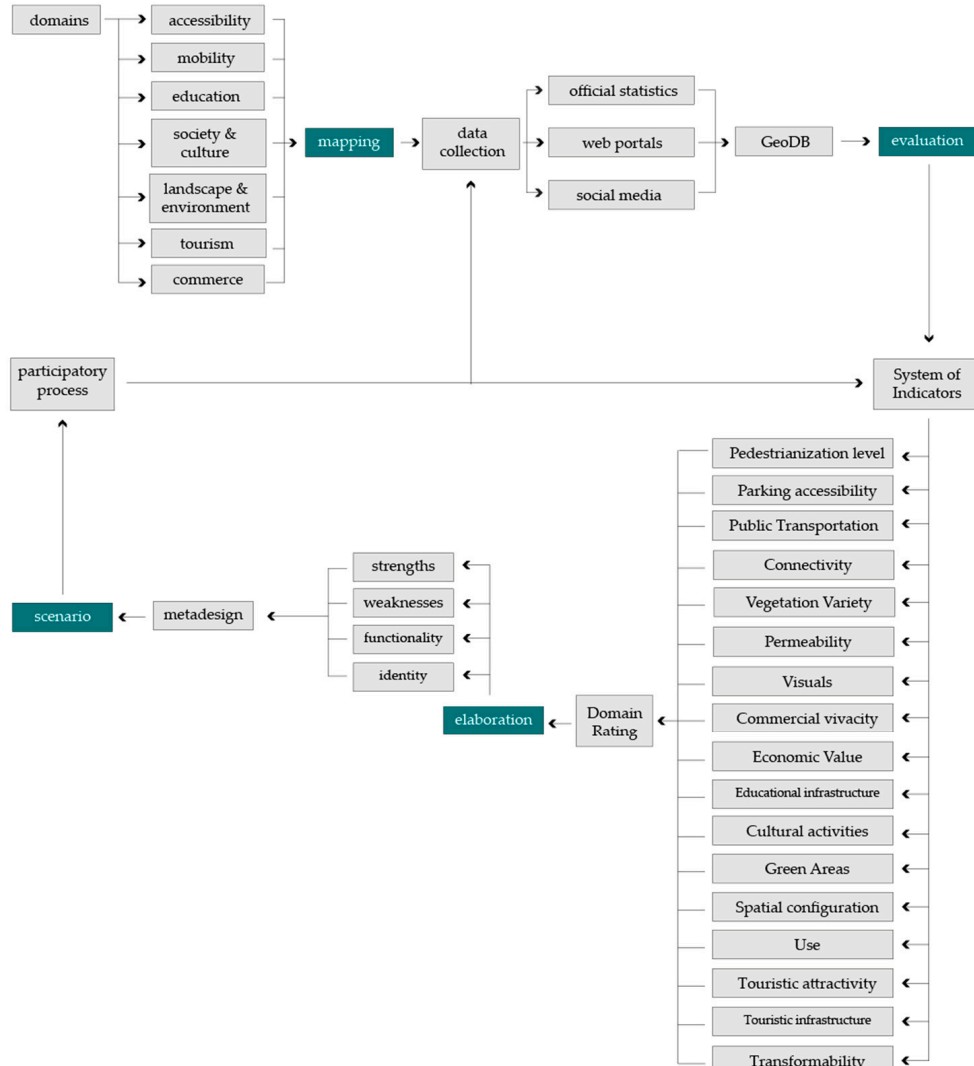

**Figure 3.** Schema of the workflow.

The main goal of these steps is to provide an analytical reading of the public spaces (mapping) from different disciplinary perspectives, integrating the results in a unique evaluative synthesis (evaluation) of the main characteristics of each place and the squares as a system. The outcomes work as input for developing strategic guidelines (elaboration), setting decision priorities, and providing development scenarios. The workflow is intended as an open framework and participatory process: collecting new data or suggesting new indicators modify the evaluative synthesis, and consequently, decision priority and development scenarios.

### 2.2.2. Mapping

A map, a diagram, or more generally, a spatial representation, helps develop structural knowledge [47] that overcomes the limitations of human perception.

In urban studies, it has been possible to realize qualitative maps of the urban environment, based on quantitative indicators (300.000 km/s Barcelona's inequality, livability, walkability) [33], to visualize and explore human–environment connections that were previously invisible [48], unveiling the digital traces [49] left by ordinary or temporary city users who share content on location-based communication platforms [50].

Indeed, the datasphere generates another level of complexity for the urban environment, and it represents a source of information on how even ephemeral and temporary phenomena can impact more tangible components, becoming a base of knowledge to predict and design development scenarios in a rapidly changing ecosystem.

The integration of Geographic Information Systems' (GIS) tools and methods in our workflow supported the multidisciplinary exploration of the squares as a system of interrelated, tangible and intangible layers and components, providing the technical framework for the collection, storage, elaboration, and visualization of data from several different sources, both institutional and not institutional (Table 1).

**Table 1.** The table presents the data-sources in detail.

| Name | Source | Datatype | Format | Data Category |
|---|---|---|---|---|
| ISTAT | Institutional | Tables, Polygons | CSV, SHP [1] | Demography, Buildings |
| Open Street Map | Open API | Points | GeoJSON [2] | Amenities |
| Virgilio | Web portal | Points | GeoJSON | Economic activities |
| Immobiliare | Web portal | Points | GeoJSON | Real estate |
| Tripadvisor | Web portal | Points, texts, images | GeoJSON | Tourism, economic activities |
| Instagram | Social media | Points, images, texts | GeoJSON | Perception and use |
| Foursquare | Social media | Points | GeoJSON | Economic activities, perception and use |
| Flickr | Social media | Points, texts, images | GeoJSON | Perception and use |
| Copernicus [4] | Remote sensing | Raster | GeoTIFF [3] | Vegetation |
| Regional geoportal [5] | Geoportal | Geometries | SHP | Basemap |

[1] Shapefile format (SHP) or [2] GeoJSON are standard formats designed for representing geographical features; [3] GeoTIFF is a standard format for georeferenced raster data; [4] the spatial resolution of the data is 10 mt.; [5] the scale of the drawing is 1:5000.

This heterogeneity required ad hoc strategies to mine data, harmonize their formats, and link them to spatial features with a common spatial reference system.

They collected demographic and building data related to the municipality of Alessandria from the National Institute of Statistics (ISTAT), joining them to the territorial bases of the census tracts. We have mined the Open API of Open Street Map (OSM), querying the existing amenities in the administrative boundary of Alessandria. The data collection was also focused on the economic vivacity of the area, scraping thematic web portals through Python algorithms. From Virgilio the economic activity data were collected, converting their address in point features through the OSM geocoding API (Nominatim). From the web-portal Immobiliare, the city's selling and renting advertisements, which provided related prices and characteristics, were downloaded and geolocated as points using the associated WGS84 coordinates. From Tripadvisor, the hotel, restaurant, and attractions data for the Alessandria district were taken, and were provided with qualitative (rating, ranking, number of reviews) and quantitative (the content of reviews, categories) attributes, and geolocated as punctual geometries through WGS84 coordinates.

In addition, referring to some experiments in the urban analysis [51,52] that work on Location-Based Social Media (LBSM) data to investigate the perception and use of public spaces of both local citizens and temporary communities, Instagram, Foursquare, and Flickr were included as sources.

The Foursquare API was helpful for the research in finding existing Foursquare places in the municipal territory, which includes most of the commercial activities and public services. Thus, this layer enriches the socio-economic information, providing a quantitative parameter (number of check-ins) to measure popularity, in relation to the community of Foursquare users [53].

Flickr and Instagram are social media which are mainly used for sharing visual content (images). Nevertheless, they serve different targets (and use different geolocation criteria): the first has a smaller community of users but it offers the possibility of sharing photos in different formats, with a high percentage of professional photographers or tourists using their trip albums. In this case, if the shooting device has the geolocation enabled, the image maintains, in its metadata, the coordinates. Thus, after querying the API, the points in the city where a Flickr photo was taken and shared were downloaded, along with other metadata (such as the timestamp, hashtags, the description, etc.).

Instagram connects a broader and younger community: it counts around a billion active users, among which 70% are under 35 years old. Thus, it permits more precise and updated information on trends concerning popular places in the city or even temporary Points of Interest (POI). In addition, Instagram geolocation works on the geotagging system: the images do not have coordinates, but the user can tag the shared photo in an Instagram place. Thus, from an urban analysis perspective, the distribution of Instagram places in the city, the number of pictures per place, and the visual characteristics of the posts become significant information. Even if Instagram has stricter API regulations, the use of scraping techniques is widely accepted in the literature. In this sense, a Python algorithm has collected a dataset with the existing Instagram places in Alessandria, with the quantitative attribute of the number of geotagged photos [54] (Figure 4).

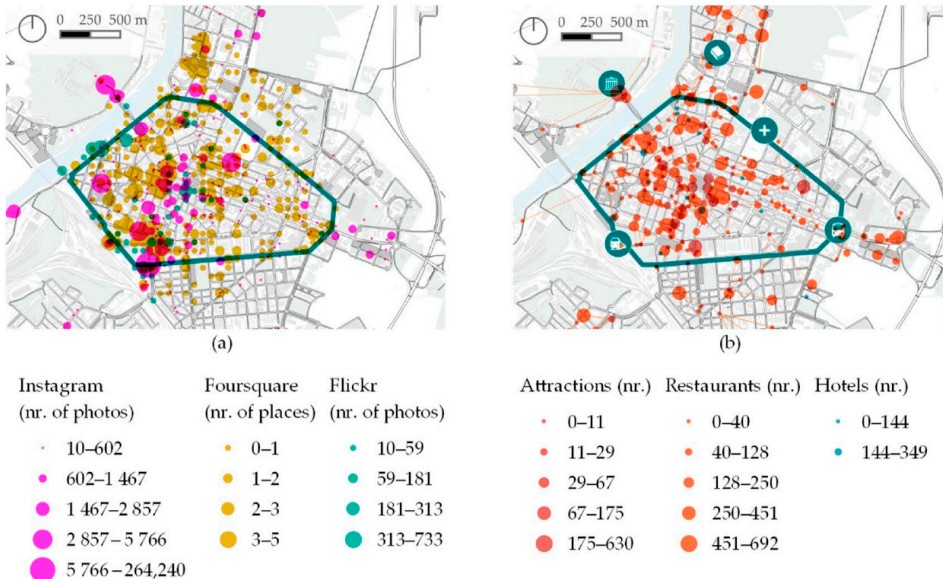

**Figure 4.** (**a**) The map shows the distribution of social network activity in the historical center of Alessandria: in magenta, the Instagram places, sized by the number of photos, in yellow, the different concentrations of Flickr photos, in green, the different concentrations of Foursquare places; (**b**) the map shows the distribution of digital activity related to touristic attraction; in particular, in orange, the restaurants, sized by the number of Tripadvisor reviews, in red the attractions, sized by the number of reviews, in green, the hotels, sized by the number of reviews, whereas the lines link the restaurants/hotels to the closest attraction [Sources: Authors, OpenStreetMap, Municipality of Alessandria].

This corpus of knowledge has been organized in an SQLite relational geodatabase (DB): it is a format selected for its portability, which permits one to have an SQL database in one file, its interoperability, its ability to be manageable with both opensource and proprietary GIS software, and its ability to perform spatial analysis and to share information among the different stakeholders.

The DB is organized in 23 tables (Table 2), where the data from the sources mentioned above have been stored, thus harmonizing their cartographic projection (WGS84-UTM33).

**Table 2.** The table presents the structure of the geodatabase.

| Layer | Type | Source |
|---|---|---|
| Interior accesses | Point | Regional geoportal |
| Real estate adds | Point | Immobiliare |
| Green areas | Polygon | Regional geoportal |
| Tripadvisor attractions | Point | Tripadvisor |
| Virgilio activities | Point | Virgilio |
| Woods | Polygon | Regional geoportal |
| Energetic cadastre | Polygon | Informative System for the energetic performance of buildings (SIPAE) |
| Waterways | Polygon | Regional geoportal |
| Regional technical map (CTR) | Lines | Municipality |
| Buildings | Polygons | Regional geoportal |
| Railway | Lines | Regional geoportal |
| Flickr photos | Points | Flickr |
| Hotels | Point | Tripadvisor |
| Worship places | Polygon | Regional geoportal |
| Civic number (ext. access) | Points | Regional geoportal |
| Parking areas | Polygons | Regional geoportal |
| Foursquare places | Points | Foursquare |
| Instagram places | Points | Instagram |
| ISTAT buildings and population census | Table | ISTAT |
| Restaurants | Points | Tripadvisor |
| Schools | Points | MIUR |
| Streets | Lines | Regional geoportal |
| Streets | Polygons | Regional geoportal |

Table 2 contains data related to the whole municipal territory of Alessandria. Nevertheless, to link this information to our object of study (the network of 13 squares in the historical center), further geoprocessing operations have been performed.

It was thought that the squares as urban components were particularly active at different scales: in the broader context of the historical center, they work as an interconnected system; in terms of the bigger scale of their surroundings, considering a walkable distance of around 5 min, we have defined a buffer of 250 m (radius from the centroid) to include in the analysis, so that the space is not strictly delimited by the perimeter of the square, and that it includes the nearby context. The selected size of the radius stands on the distance that, in the existing literature on sustainable cities, is considered ideal for pedestrian movement [55–57].

It has taken the use of wider geometry as a basis to link the data to the squares through spatial join, achieving, as a result, a new, unified table with one record per square and

23 attributes with data on its physical structure, socio-economic and demographic dynamics, prevalent activities and services, perception, and ordinary or temporary uses (Figure 5).

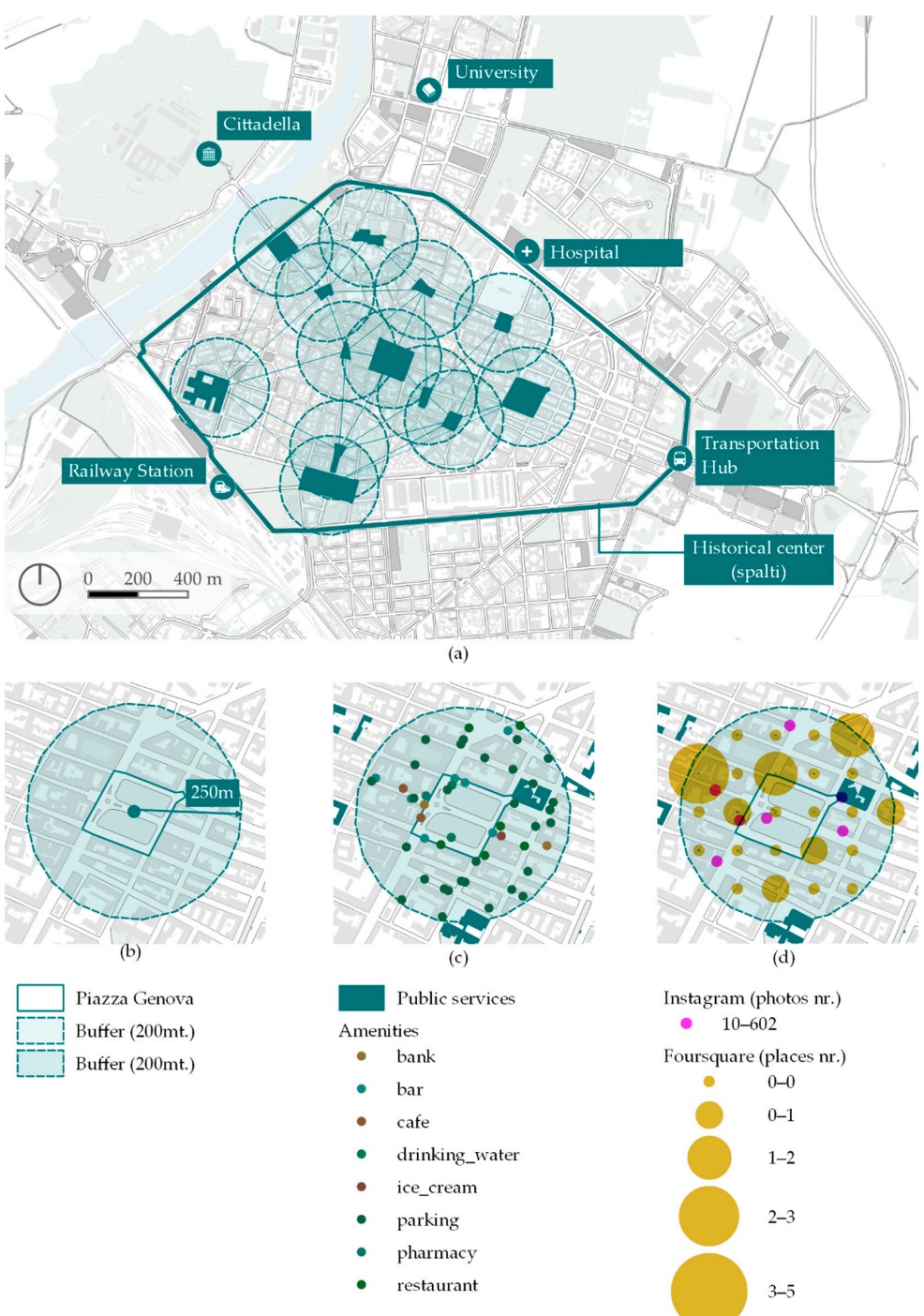

**Figure 5.** (**a**) For each of the selected squares, a buffer has been created, (**b**) calculated using a 250 m radius from the centroid of the perimetral polygon, as shown in detail in the example of Piazza Genova/Matteotti; (**c**) this geometry has been used as a basis for collecting data related to the square and its area of influence, such as amenities and public services marked in the picture; (**d**) and for revealing its use through social media datasets; for instance, in the picture, the magenta dots are the Instagram places (sized by a number of photos), the yellow ones are the concentration of Flickr photos, whereas the green ones are the concentration of Foursquare places [Sources: Authors, OpenStreetMap, Municipality of Alessandria].

In addition to measuring the physical space and exploring the virtual dimension of the datasphere, we included the space syntax segment and isovist analysis in the mapping action. These methods analyze the spatial configuration of the urban space returning indexes to measure how it influences the movement and perception of human agents.

2.2.3. Evaluation

In the evaluation phase, the research team converted this information system into an evaluative one, aggregating and elaborating data as indicators to rate the strengths and weaknesses of each square, with regard to the predefined domains.

To do so, a synthetic evaluative abacus has been used as a mediation tool. Its structure contains a column per square, a first level row per domain, and for each domain, a sublevel per indicator. Each indicator consists of a synthetic name, a general description, a definition of clear and replicable evaluation criteria, and a score (Appendix A). The score ranges from one to five points per indicator

Using the geodatabase as an input and the abacus as a reference, the domains were interpreted through different disciplinary perspectives, performing geospatial analysis through GIS tools, aggregating data and proposing indicators provided by specific evaluation criteria and the related score (Figure 6).

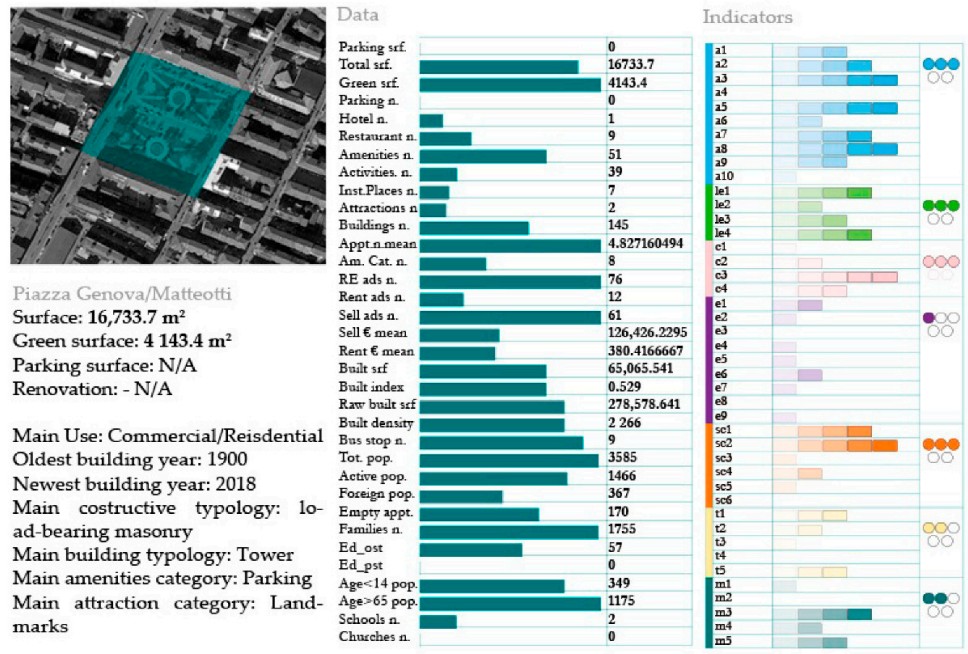

**Figure 6.** Taking Piazza Genova/Matteotti as an example, the collected data have been aggregated as a thematic set of indicators, moving from the first mapping phase to the evaluative one, and laying a foundation for subsequent strategic design elaboration [Source: Authors, Google Satellite].

The accessibility topic has been interpreted through four main perspectives: pedestrian accessibility of the square, functional heterogeneity, services for fragile categories, and position about the street graph.

The indicators for the landscape and environment domain follow three main interpretative lines: (1) the evaluation of the botanic and vegetational quality and variety, considering diversity as a quality of the green areas; (2) the perceptive dimension that includes, in a holistic sense, the architectonic quality of the buildings in the surroundings, with the presence of some recognizable sightlines or some outstanding vegetational element; (3) the transformability of the squares, with a focus on a sustainable interpretation, evaluating the possibility to change the pavements for draining and recycle the rainwater.

The commerce domain has been analyzed through indicators that measure the area's economic value and business vivacity and through parameters that are more focused

on potential development. In particular, the massive presence of parking areas has been considered as a negative factor for commercial attractiveness, given the inverse proportional correlation with the number of photos (Flickr and Instagram), in relation to the squares that have a higher percentage of parking areas.

The indicators related to the educational domain included both parameters to measure the existing infrastructures and services, and the transformability of the squares, considering the potential users and the physical spatial configuration.

In the analysis of the social and cultural components, they have been integrated in an evaluation of the cultural attractiveness of the square, which is intended to show the presence of related places and services, and the perception and use of the public spaces that are revealed by social media activity.

For tourism, the level of attractiveness and touristic infrastructure has been measured, with a specific focus on the relationship between hotels and restaurants. Furthermore, for mobility, the possibility of converting the actual level of interconnection of the squares into intermodal exchange nodes has been considered.

Finally, to achieve a synthetic evaluation, the indicator scores have been integrated, through a system of weights, using a domain-based rating.

A digitalized version of the populated abacus updated the squares table in the geodatabase, with the new attributes related to the indicator scores and the domain rating, which provides the basis for visualizing a synthetic multidimensional evaluation for each square (Figure 7).

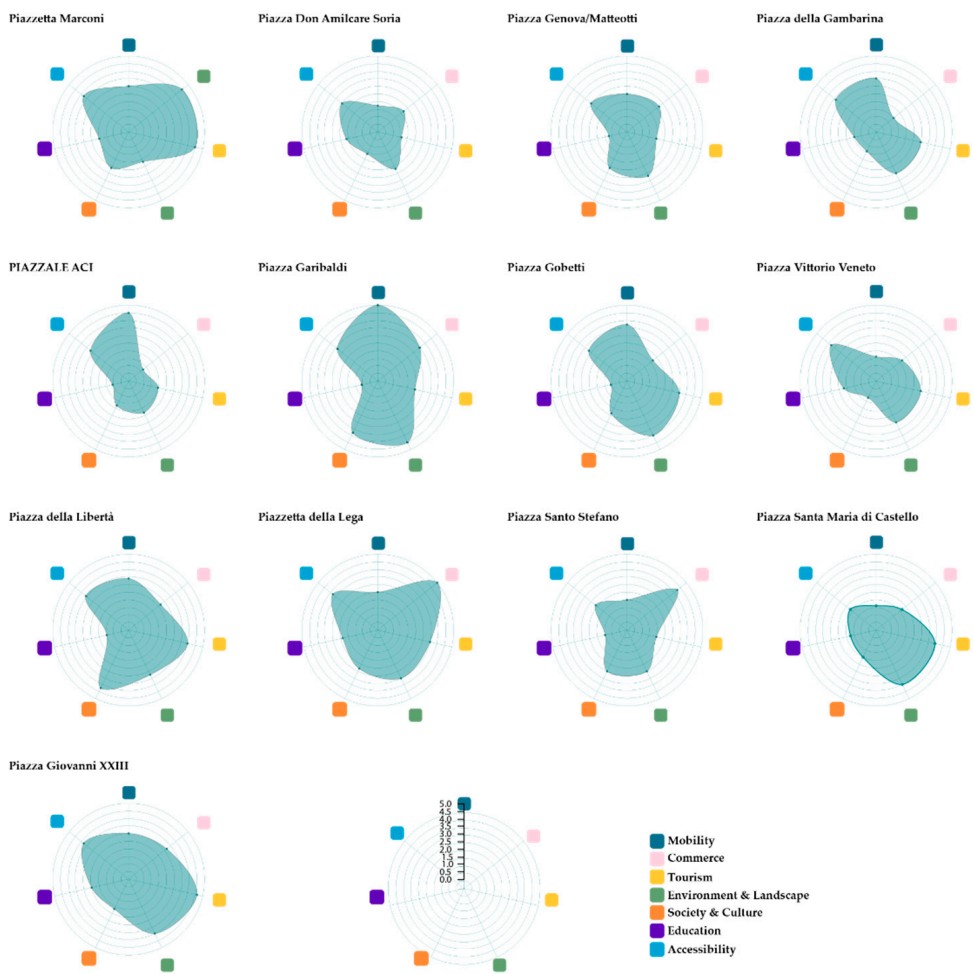

**Figure 7.** The polar diagrams of the 13 squares in order to visualize the synthetic domain rating [Source: Authors].

This data visualization outcome is not secondary to the MCDA workflow, offering the opportunity to provide decision-makers with a transparent, sharable, and updatable information system for setting priorities, as well as a dialogic tool that represents each square in its specificity, in accordance with the deeper analytic levels of the parameters and the domain-related synthesis, and in terms of weaknesses and transformative potential.

The main achievement of this phase of the workflow was to define parameters that were not necessarily objective, being subjectively dependent on the ratio of the evaluation criteria, but scientific, in the sense that it can be replicable and coherent with the general framework of MCDA.

### 2.2.4. Elaboration

The passage to the next step of the workflow was also simulated, moving from the analysis and the evaluative synthesis to the elaboration of a development strategy.

Indeed, considering the strengths, weaknesses, and specificity of each square, and shifting between the two scales of the squares network and the individual dimension of each place, a first draft of the meta-design proposal has been developed (Figure 8).

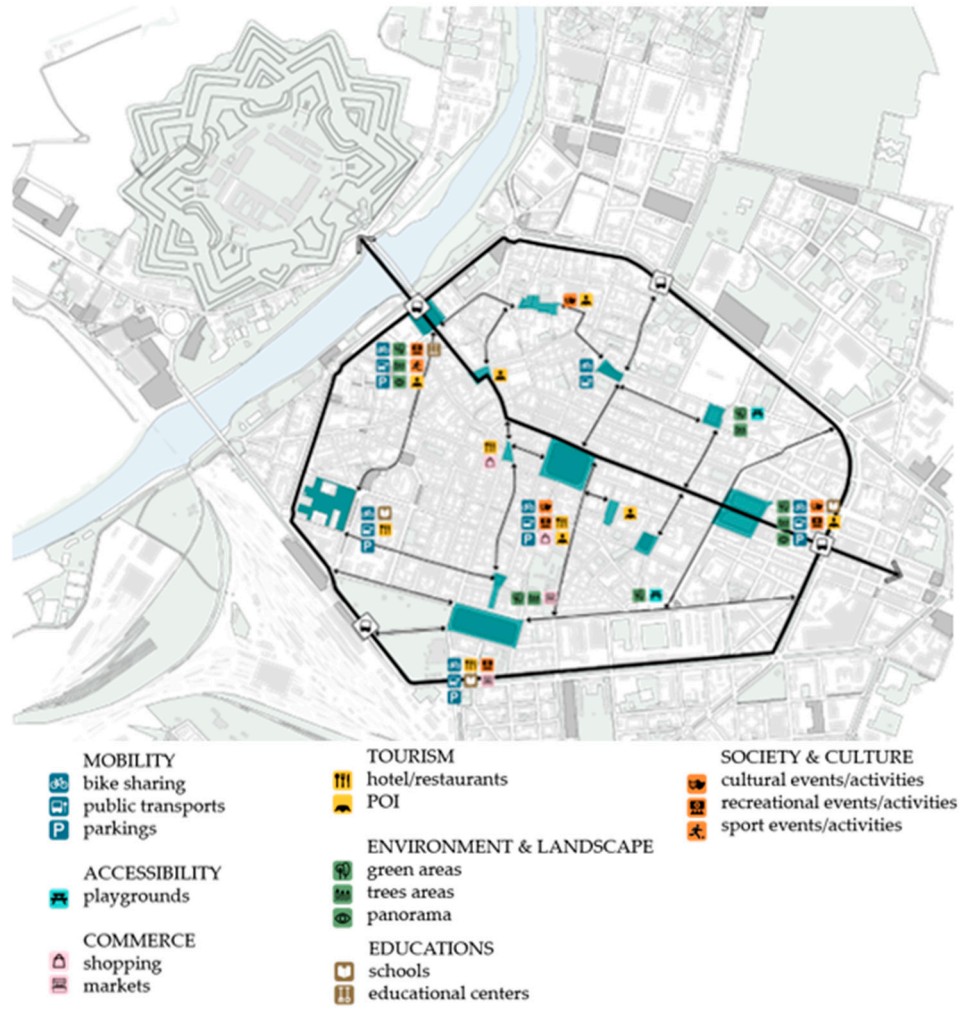

**Figure 8.** The proposed meta-design with the East–West backbone, the network of pathways that reconnects the squares, and the proposed functions [Sources: Authors, OpenStreetMap, Municipality of Alessandria].

At the core of this hypothesis are the natural resources, which have been taken advantage of, that emerged from the evaluation of the landscape and environmental domain, and

which can be considered as a systemic vector of development, with beneficial effects on other components of the squares, and more generally, on the quality of the historical center.

The first reason for this choice is due to the double physical and symbolic sides of the main natural elements of the city, the river, which works as a boundary and a connection, representing, at the same time, a resource and a threat.

The closeness between the city and the Tanaro river makes its riverside, with a few punctual interventions, the most suitable place for integrating the scarcity of open spaces in the historical center. This process requires a rethinking of the complex and conflictual relationship between the city and the water and a recovery of its original positive role through a transformation of how the water invades the urban space: not as a menace but as a connection. A sound, an echo, a sign through which a natural element marks a hierarchy in the street network, orienting the movement of people.

Indeed, the visibility indicators generated by space syntax analysis, and the qualitative evaluation of sightlines, lets a West–East axis emerge that crosses the city center, reconnecting, on a smaller scale, the Western side of the river Tanaro, and the monumental fortress of the Cittadella, with the oriental, industrial, and agricultural, part of the municipal territory, marked by the path of the river Bormida.

The ideal conjunction of these natural elements allows the water to enter the city center, as a minimal urban sign, through the gate of Piazza Gobetti (that reclaims the original name of Piazza Tanaro), qualifying the backbone of a network of mixed or pedestrian streets that links the 13 squares.

This orientation line proceeds to the institutional core of the city (Piazza della Libertà), and following the street of Via Dante, renovated in its role of the commercial axis, it arrives at the Eastern gate of the central area, Piazza Genova, where the renovation of the historical gardens and the valorization of the monumental Arch of Marengo become the core of a development scenario.

A second reason that has stressed the role of environmental and landscape indicators in decision prioritizing concerns, is their cascade effects on the general objective of sustainable development.

The enhancement of the quality and percentual quantity of green areas in public spaces prompts walkability indicators to arise, naturally encouraging pedestrian mobility, with beneficial effects on people's health and pollution reduction. Furthermore, the vegetation permits the generation of reactive strategies to the UHI phenomena, with benefits for the accessibility of the squares and the optimization of energy consumption.

On this basis, after elaborating the indicators in a general meta-design for the system of squares, the students have hypothesized a transformation scenario for the two squares/gates (Piazza Gobetti and Piazza Genova), where environmental issues have a higher impact.

### 2.2.5. Scenario

As an open conclusion of the paper, the research team proposed the renovation of Piazza Genova: the square with the highest percentage of green areas. The valorization of natural elements represented a specific objective for the PA, in terms of which systemic effects have been recognized in the synthetic evaluation.

Piazza Genova is a 16,733 m$^2$ square, located in the Eastern area of the city center, representing an extra-urban access point (Figure 9).

Its main characteristics consist of the presence of an undervalued cultural heritage element (the Arch of Marengo) and a historic high school. Moreover, given the connection with the commercial axis of Via Dante and with the expressway of the *spalti*, the gardens, in the general trend of squares mainly used as parking areas, represent a resource to be valorized (Figure 10).

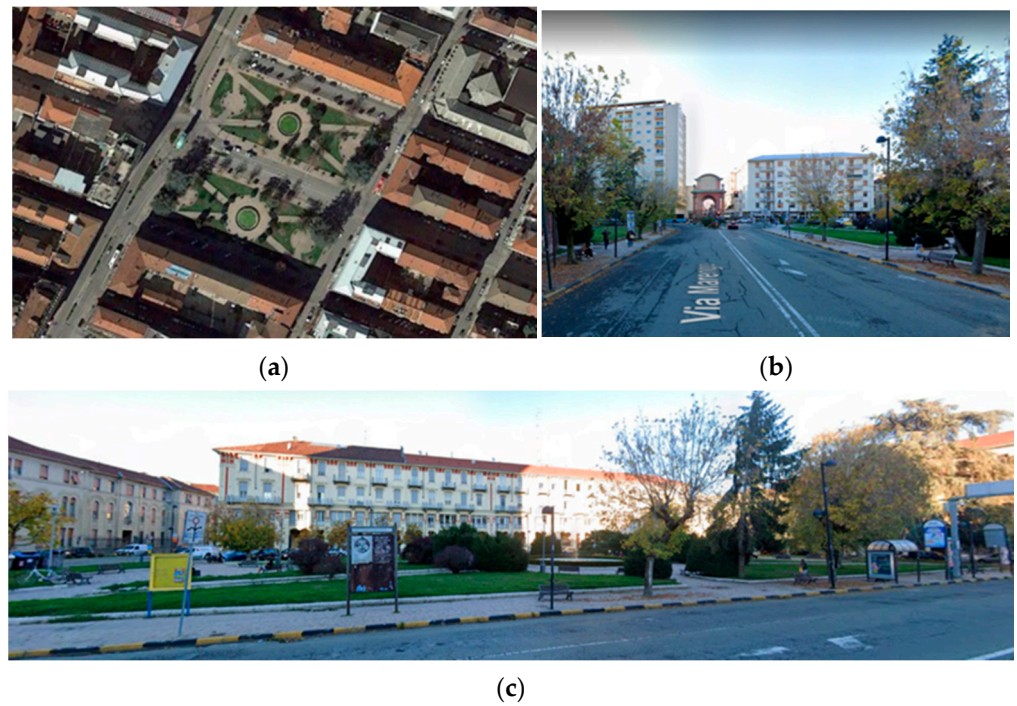

**Figure 9.** The photos represent Piazza Genova/Matteotti (**a**) from an aerial perspective, showing the structure of the two small parks on the sides of the central axis of Via Marengo (**b**), that continues in the direction of Via Dante, from the historical center to the extra-urban areas, starting from the monumental Arch of Marengo; (**c**) from a human-scale perspective, the borders of the parks are lowly densified, mainly defined by small size trees like *Acer Negundus* (Source: Google Street View).

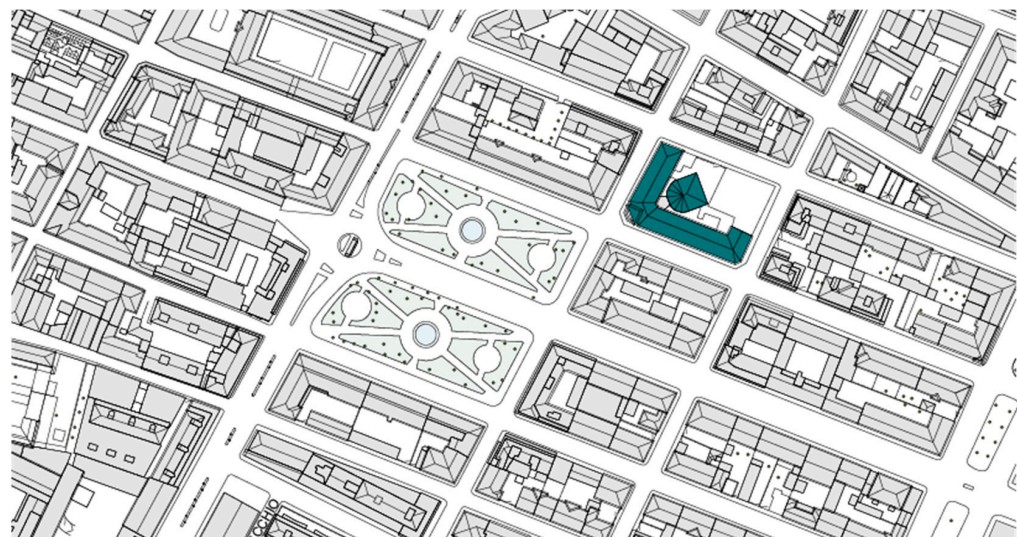

**Figure 10.** The technical map of Piazza Genova gives a more precise idea of the present configuration of the square; it is divided into two mirrored small parks, with two central fountains, introduced by the Arch of Marengo on the Western side and connected to the building of the high school Liceo Classico "Giovanni Plana" (in blue); in particular, the distribution of the trees follows a circular path around the central points of the parks, marking the ideal axis that reconnects the two fountains [Source: Municipality of Alessandria].

In the evaluation phase, Piazza Genova showed potentialities related to accessibility, due to its strategic position on the perimeter of the *spalti*, being at the Eastern gate of the city center, and due to the environmental domain, given the presence of green areas. On

the other hand, there are critical points concerned with the reconnection of the nearby high school with the activation of cultural spaces and activities that, in general, would give the square a specific identity. Based on this premise, in the elaboration phase, they have been defined as possible strategies to enhance the role of Piazza Genova in the system of squares as a multimodal interchange node: a connective function harmonized with the new identity of "square of the arts", with a specific identity given by the requalification of the parks. The development scenario is based on a main reconnective action that involves the square and its components at different scales (Figure 11).

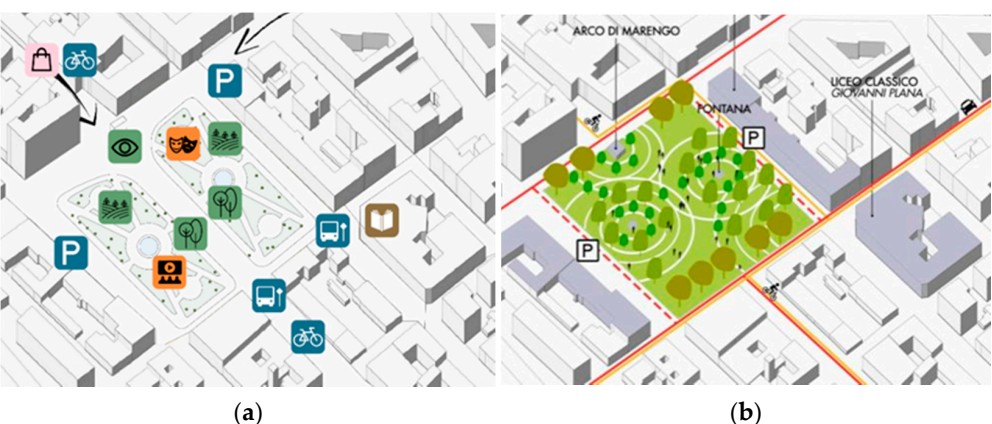

(**a**)                                                                                (**b**)

**Figure 11.** (**a**) The schema brings the strategies proposed in the elaboration phase to the square scale, focusing in particular on the locations of service parking areas and the configuration of the square as an intermodal mobility hub, and on the requalification of the parks, considering three main aspects: the quality of the sightlines, the enhancement of the vegetational elements and the improvement of permeable surfaces, and on the activation of spaces for cultural events; (**b**) the strategies mentioned above are turned into a proposal for a transformation scenario [Source: Authors].

As seen in the elaboration phase, Piazza Genova becomes an intermodal exchange node that provides the main access to the historical center, enforcing the commercial role of Via Dante. At the square scale, the two sides of the garden (now divided by a central vehicular street) are joined in a unique redesigned park. At the side-street scale, the permeable borders of the new park include the nearby covered walkways, avoiding the fragmentation and isolation that has caused some commercial activities to close.

The Arch of Marengo recovers the original role of the landmark for the walkable pathway from/to the city center, and it marks the entrance of the park. The green space is conceived as a system of more intimate places, divided by vegetational elements, where it is possible to organize cultural events or to host, in connection with the school, the student community.

## 3. Results

The results of the synthetic evaluation provided by the expert-based SI presented in this article, has revealed three main recurrent patterns, identifying three typologies of squares, and consequently, it has allowed the development of ad hoc design strategies:

- big marginal squares, with a prevalence of parking areas: highly accessible and interconnected, but not attractive for economic and social activities, with a functional role for mobility, but with a less defined specific identity;
- small squares, hidden in the street network of the historical center, sometimes recently renovated, with a high potentiality attracting tourism or social aggregation, but underused, due to their lack of interconnection and visibility;
- symbolic squares, with a generally high score in the socio-cultural domain and high quality of the architectural environment, but without outstanding attractiveness, maintaining a principally functional role for parking areas.

On this basis, a development strategy was defined based on two core actions:

1. reconnection of the squares in a system of pedestrian and perceptive pathways, emphasizing the specific identity of each square and their functional role in the system;
2. enhancement of the environmental quality of the squares, introducing and valorizing the vegetational elements that work both as place identity-makers and as drivers for beneficial influences on human healthiness.

This strategy, considering the cascade effects on other indicators, prioritizes the actions related to the landscape and environmental domain, focusing in particular on:

- the redefinition of the presence and perception of water in the urban environment, from menace to guide as part of the pathways system, with a positive impact on the visibility of the small squares and on the renovation of Piazza Gobetti, which is reinterpreted as the river-gate;
- the enhancement of the vegetational quantity, quality, and diversity of green areas, paying specific attention to their aesthetic interaction with the built environment, in terms of sightlines and perspectives, with a positive impact on the social and economic attractiveness of the squares;
- the sustainable development of the squares, considering the role of vegetational elements in mitigating the urban microclimate and the possibility to activate water recycling systems.

The participatory process will move on from the results of the synthetic evaluation provided by the first expert-based SI. The proposed workflow has provided dialogic inputs for the next participatory phase that will involve the research team and the PA. The SI forms the basis for integrating new parameters, according to the instances of the stakeholders, adjusting, consequently, the evaluative synthesis and decision priorities.

## 4. Discussion

These results have provided an operative overview of the main characteristics of the squares, revealing strengths and weaknesses, and formulating a shared base of knowledge to activate the next interlocutory phase. Nevertheless, the expert-based nature of the SI requires further validation and formalization.

However, the first hypothesis of regeneration strategies (the scenario and the draft of the masterplan) and the visual outcomes of the SI (maps and graphs) are permitted to fill the gap between the abstractness of the evaluation system (and of its domains and indicators) and the decision-making requirements of the PA.

In particular, the paper proposes an advancement in the direction of developing methods for collaborative decision-making processes in urban planning, exploring the application of MCDA as a mediation tool [58].

The proposed solution stands on the idea that the effectiveness of MCDA in decision-making processes related to urban planning is enhanced by the integration of heterogeneous input, which supports the definition of different criteria, and consequently, permits the incorporation of the value sets of different stakeholders [59,60]. In particular, the integration of LBSM datasets follows the direction of collaborative MCDA [61], encouraging the involvement of citizens, managers, and experts.

Furthermore, the adopted workflow includes spatial visualization (maps and diagrams) as a process to reduce cognitive bias in MCDA, improving the mediation role of SIs [62]. In addition, following this direction, the SI has been intended not as a fixed but a flexible tool, open to be transformed and adapted in the decision-making process, but maintaining the fundamental role of dialectic synthesis.

In the interlocutory phase, it will be necessary to pay attention to the bias inscribed in selecting the data sources, and to not to drive to conclusions that are not related to the specificity of the collected dataset [63]; however, in the following phases, it will be possible to eventually integrate the corpus through new data collection processes [64].

In parallel with the participatory phase, the research plan aims to implement the SI in two ways:

- aggregation of indicators as new, more formalized ones, to measure, in particular, the walkability of the squares, as an inclusive parameter that integrates lived, perceived, and physical environments;
- introduction of specific indicators for monitoring the vegetation quality, variety, and temporal evolution, including new remote sensing data to calculate vegetational indexes.

However, this formalization of the first SI will keep the original and structural heterogeneity in the general framework of a mapping process that tries to synthesize, not simplify, the complex nature of public spaces and the urban environment.

## 5. Conclusions

In their intimate nature, public spaces are used spaces that are crossed, seen, and lived in by resident and external communities of people. Thus, even if the spatial structure of these spaces determines the constraints and opportunities for human actions, how they are used represents a significant input for urban planning, revealing hidden points of interests, pathways, and potentialities.

The digital tools, methods, and mapping techniques permit these invisible layers to be revealed, laying the foundation to provide new inputs for decision making. In this sense, the integration of qualitative data and information, particularly from social media, allows the addition of a transformative perspective of what a square has been and could be, based on the temporality of the different ways people use and perceive them.

Furthermore, the elaboration of data in indicators, and their integration in a synthetic evaluative system, builds a bridge between the descriptive phase and the operative decision making requirements.

In this sense, the SI tool works more as a mediator, providing a knowledge base for generating dialogue, rather than a modelling system. The definition of a shared and clear methodological framework makes it possible for the different stakeholders to contribute to decision-making effectively.

**Author Contributions:** The research was conceived and developed by all three authors, and, more specifically: A.B.: conceptualization and methodology; H.N., M.V.: validation, formal analysis, investigation, resources, and data curation; H.N., M.V. writing—original draft preparation; H.N., M.V.: writing—review and editing, A.B.: visualization, supervision, project administration, funding acquisition. All authors have read and agreed to the published version of the manuscript.

**Funding:** Funding by MIUR (Ministry of University and Research) 2020 Principal Investigator Prof. Alessandra Battisti, collaborators H. Natta e M. Valese. The research "Città pubblica e nuovo welfare | Public city and new welfeare". was funded by MIUR 2020 RM120172B3B15DB9 for the basic research activities of Prof. Alessandra Battisti.

**Acknowledgments:** The research is funded and developed in the institutional framework of cooperation between the University of Rome—La Sapienza and the Municipality of Alessandria.

**Conflicts of Interest:** The authors declare no conflict of interest. The funders had no role in the design of the study; in the collection, analyses, or interpretation of data; in the writing of the manuscript, or in the decision to publish the results.

## Appendix A

**Table A1.** The table presents the SIs in detail, with a description of the indicators and the related evaluation criteria.

| Domain | Indicators | Description | Evaluation Criteria |
|---|---|---|---|
| Accessibility | a1. Pedestrianization level | Pedestrian percentage of square surface | S = 1: 0%; S = 5: 100% |
| | a2. Parking accessibility | Distance (in minutes) from parking | S = 1: 30 min; S = 5: 5 min |
| | a3. Urban public transportation connectivity | Number of urban public transportation lines that cross the square | S = 1: 0; S = 5: 4+ |
| | a4. Extra-urban transportation connectivity | Number of extra-urban public transportation lines that cross the square | S = 1: 0; S = 5: 4+ |
| | a5. Public transportation accessibility | Distance (in minutes) from bus stop | S = 1: 30 min; S = 5: 5 min |
| | a6. Services for fragile categories | Number of services in the 250 m buffer | S = 1: 0; S = 5: 4+ |
| | a7. Centrality | Centrality index calculated on the basis of the number of streets (links) to which a square (node) is connected | S = 1: i < 0.0005; S = 2: 0.0005 < i < 0.0006; S = 3: 0.0006 < i < 0.0007; P = 4: 0.0007 < i < 0.0008; S = 5: I > 0.0008 |
| | a8. Closeness | Mean distance on the graph between the square and the other nodes | S =1: d < 7400; S = 2: 7400 < d < 7500; S = 3: 7500 < d < 7600; S = 3: 7600 < d < 7700; S = 3: 7700 < d < 7800; S = 5: d > 7800 |
| | a9. Betweeness | Number of times that the square has the shortest path between other nodes on the graph | S = 1: n < 10,000; S = 2: 10,000 < d < 20,000; S = 3: 20,000 < d < 50,000; S = 3: 50,000 < d < 100,000; S = 3: 100,000 < d < 200,000; S = 5: d > 200,000 |
| | a10. Heterogeneity | Number of different OSM amenity categories in the 250 m buffer | S = 1: $n < 10$; S = 2: $10 < n < 15$; S = 3: $15 < n < 20$; S = 4: $20 < n < 25$; S = 5: $n > 25$ |
| Landscape and environment | le1. Index of vegetational variety | Qualitative parameter based on the variety of the vegetation and on its typology (trees, bushes, grass) | Qualitative criteria: a higher variety corresponds to a higher score |
| | le2. Permeable surfaces | Ratio between green surface/total surface of the square | S = 0: percentage = 0%; S = 5: percentage = 100% |
| | le3. Potential water recyclability | Impermeable surface: Inverse ratio between green surface/total surface | S = 0: percentage = 0%; S = 5: percentage = 100% |
| | le4. Visuals | Qualitative parameters that consider the architectural quality of the buildings on the squares, the quality of sightlines and gates, and the presence of vegetational elements | |

| | | | |
|---|---|---|---|
| Commerce | c1. Commercial vivacity | Number of economic activities (Virgilio) in the 250 m buffer. | S = 1: $n < 40$; P = 2: $40 < n < 50$; P = 3: $50 < n < 111$; P = 4: $111 < n < 130$; P = 5: $n > 130$ |
| | c2. Food and drink sector vivacity | Number of Tripadvisor restaurants | S = 1: $n < 9$; S = 2: $9 < n < =10$; S = 3: $10 < n < 15$; S = 4: $15 < n < 20$; S = 5: $n > 20$ |
| | c3. Economic value indicator | Mean of apartment selling prices (Immobiliare) in the 250 m buffer | S = 1: $p <$ EUR 90,000; S = 2: $90,000 < p <$ EUR 110,000; S = 3: $110,000 < p <$ EUR 130,000; S = 4: $130.000 < p <$ EUR 190,000; S = 5: $p >$ EUR 190,000 |
| | c4. Parking presence | Ratio between the parking area and the total surface of the square | S = 1: 0%; S = 5: 100% |
| Education | e1. Educational infrastructure level (surroundings) | Number of schools in the 250 m buffer | S = 1: $n = 0$; S = 5: $n = 4+$ |
| | e2. Educational infrastructure level (square) | Number of schools on the square | S = 1: $n = 0$; S = 5: $n = 2+$ |
| | e3. Cultural activities indicator | Number of services and cultural associations on the square | S = 1: $n = 0$; S = 5: $n = 3+$ |
| | e4. Cultural vivacity of the square | Number of recreational activities on the square | S = 1: $n = 0$; S = 5: $n = 3+$ |
| | e5. Pedestrian infrastructure | Percentage of pedestrians on the whole of the square's surface | S = 1: 0%; S = 5: 100% |
| | e6. Green areas | Percentage of green surfaces out of the whole of the square | S = 1: 0%; S = 5: 100% |
| | e7. Thematic itineraries | Number of thematic itineraries that cross the square | S = 1: $n = 0$; S = 5: $n = 5+$ |
| | e8. Potential development of integrative activities | Qualitative parameter based on the evaluation of the square surface, the existing activities/services, and the buildings. | |
| | e9. Students | Number of resident students in the 250 m buffer | S = 1: $10 < n < 20$; S = 2: $20 < n < 30$; P = 3: $30 < n < 40$; S = 4: $40 < n < 50$; S = 5: $n > 50$ |
| Society and culture | sc1. Configuration as potentiality (sup. mq) | Total surface of the square | S = 1: 500 mq; S = 5: 25.000 mq |
| | sc2. Recreational places | Number of recreational places on the square | S = 1: $n = 0$; S = 5: $n = 5+$ |
| | sc3. Cultural places | Number of cultural places on the square | S = 1: $n = 0$; S = 5: $n = 4+$ |
| | sc4. Identitarian places | Sum of the number of photos geotagged in the Instagram places on the square | S = 1: $n = 0$; S = 5: $n = 5000+$ |
| | sc5. Cultural attractivity | Mean attraction rating (Tripadvisor) in the 250 m buffer. | S = 1: 1; S = 5:5 |

| | | | |
|---|---|---|---|
| Tourism | t1. Daytime use | Number of bars and cafés in the 250 m buffer | |
| | t2. Nighttime use | Number of restaurants, pubs, fast food outlets, and nightclubs in the 250 m buffer | |
| | t3. Attractivity of public space | Number of posts geotagged to the nearest Instagram place | S = 1: *n* < 200; S = 2: *n* < 300; S = 3: *n* < 400; S = 4: *n* < 500; S = 5: *n* = 500+ |
| | t4. Touristic infrastructure | Hotels and restaurants as ratio of eight restaurants to one hotel | S = 0: h = 0; S = 1: h >= 1 & r <= 8; S = 3: h > = 2 & r < = 16; S = 4: h > = 3 & r <=24; S = 5: h > = 4 & r < = 32 |
| | t5. Touristic preference | Mean restaurant rating (Tripadvisor) | S = 1: m < 3.8; S = 2: 3.8 < m < 3.9; S = 3: 3.9 < m < 4; S = 4: 4 < m < 4.2; S = 5: m = 4.2+ |
| Mobility | m1. Relevance for car mobility | Ratio parking area/ total surface | S = 1: r < = 20%; S = 2: 20% < r < 40%; S = 3: 40% < r < 60%; S = 4: 60% < r < 80%; S = 5: 80% < r < 100% |
| | m2. Relevance for soft mobility | Distance from cyclable path | S = 1: d > 200 m; S = 5: 0 |
| | m3. Transformability (soft mobility interchange node) | Qualitative evaluation of the position of the square in relation to the city, the street network, the main activities (OSM), the street section, and the traffic flow | S = 0: + distance from cycleways -transformable street sections, activities requiring car access; S = 5: − distance from cycleways, + transformable street sections, activities with mixed access |
| | m4. Transformability (soft/car mobility interchange node) | Qualitative evaluation of the position of the square in relation to the city, the street network, the main activities (OSM), the street section, and the traffic flow | S = 0: + distance from car/bike intersections—transformable street sections, pedestrian access activities; S = 5: + closeness to cycleways + transformable street sections, mixed access activities |
| | m5. Connectivity to the public transport network | Number of bus stops in the square | S = 0: 0; S = 1: 1; S = 2: 2; S = 3: 3; S = 4: 4; S = 5: 5+ |

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
