# Peer review of "Indicators as Mediators for Environmental Decision Making: The Case Study of Alessandria"

_land, doi:10.3390/land11050607_

Round 1

Reviewer 1 Report

Dear Authors,

Congratulations for your article, it was a great pleasure to read. Its scientifical soundness, structure and English language is very good and well-interpretable.

I have just a few minor comments:
- scan through the article for searching unnecessary definite articles and typos (e.g., line 38: fulfils -> fulfills);
- centuries are written in the article in various styles (e.g., 20th Century - XX century);
- is this stlye of page referencing after in-text citations correct?;
- a noun is missing around word 'widespread' in line 101;
- a verb is missing from line 121;
- the background maps (regarding map figures) were also edited by the authors? If not, please indicate the source;
- line 156: '"the city has had two main vocations in its history';
- line 166: points in decimal numbers and km2;
- consider rewording line 261;
- figures 4-5: a numerical legend should be included for the circles (e.g.,how many reviews belong to each circle sizes).
I have found some typos and not 100% clear sentences grammatically later, but a fast scan would filter them out one by one by you.

I wish you many citations!

Author Response

Dear reviewer

thank you for the review and the valuable advice and corrections.

Here it follows the detail of our review:

  1. Reviewer 1:
    • scan through the article for searching unnecessary definite articles and typos (e.g., line 38: fulfils -> fulfills);
      • we have performed a double check on English correctness. However, we have followed a British English standard, and maintained the proposed example (and other similar cases) in the British form ‘fulfil’. If specifically required, we can review the paper switching to US English standard;
    • centuries are written in the article in various styles (e.g., 20th Century - XX century);
      • we adopted the 20th Century form for every occurrence;
    • is this stlye of page referencing after in-text citations correct?;
      • looking to the Land template, we have maintained the current style of page referencing; however, if wrong, we can switch to another style;
    • a noun is missing around word 'widespread' in line 101;
      • we added the required noun
    • a verb is missing from line 121;
      • we checked the sentence, but it seems to be correct;
    • the background maps (regarding map figures) were also edited by the authors? If not, please indicate the source;
      • we added the source for the base maps;
    • line 156: '"the city has had two main vocations in its history';
      • the revision has been accepted;
    • line 166: points in decimal numbers and km2;
      • the revision has been accepted;
    • consider rewording line 261;
      • the revision has been accepted;
    • figures 4-5: a numerical legend should be included for the circles (e.g.,how many reviews belong to each circle sizes);
      • we added a legend with the required information.

Reviewer 2 Report

Reviewer comments

General comments:

The topic of the paper is very interesting, and the gap between academic and practical discussions on planning and design of public spaces exists. The abstract is concise. The introduction is well-written and gives a coherent background for the study. However, the authors need to refocus the aim and results of this paper towards the actual results presented in this study (indicator creation) and not stakeholder/planning system engagement. Consequently, the structure of the paper concerning Methods and Results needs some fine tuning. The scoring / weighting stage of the method needs more elaboration, and the role of stakeholders (or its absence) needs to be clarified. The weakest part of the paper is the Discussion section that completely lacks discussion with other scholars’ work. In addition, a further language and grammar check is still needed.

Detailed comments section by section:

1 Introduction

  • Lines 45-46: Consequently, their aesthetic/environmental quality has a strong impact on the general healthiness of people. Some scientific references are needed to mention to support this claim.
  • Line 66. What is the paradigm shift that is referred to here? Is it the one described in the next paragraphs? In that case, this paragraph should maybe merged with the next.
  • Lines 67 and 69: The words Century, Mathematics, Economy, Geography, and Social Sciences should not be capitalized
  • Line 75: XX century. Previously used the notation 20th century. Choose one notation and use it coherently.
  • Lines 95-99: However, the applicability of these SIs to urban planning, and the replicability of these theoretical models in different contexts, in the passage from the global scale to the local one, suffered limitations related to (1) the availability of the required data for elaborating the indicators; (2) the cultural shift, that hardly permits an unambiguous interpretation of indicators; (3) the complexity of the SIs, and the consequent difficulties in the integration with urban planning and decision-making workflows. These three claims certainly need some references from the academic literature.
  • Line 122: local PA. Please provide an explanation for the acronym PA
  • Line 122-123. the proposed SIs hardly match with the procedures of decision-makers. This claim needs a reference.
  • Since the paper concerns a spatial MCA tool, some recent scientific literature on those tools should be introduced in the Introduction section. For example the works of Davide Geneletti and Valentina Ferretti among others could be relevant for this approach, since they have done a lot of research on spatial decision-support systems in Italian context. The Introduction section is quite long, so maybe the description of the historical paradigm shifts could be made more compact in addition.

2 Materials and Methods

  • Figure 1: Maps a, b, and c lack a scale bar. The map d would benefit from a legend. The colour for the green spaces could be brighter for them to stand out from the base map. The isochrones visualisation of the 15-min city does not currently work, and it seems a bit unnecessary element in the map. Since the manuscript includes quite a lot of figures and maps, I would strongly suggest the authors to combine the maps Figure1 d and Figure 2.
  • Line 184: The adopted methodological approach for answering this task is framed.... Please elaborate what task. You should not refer to other sections in the sentences in another sections.
  • Lines 186-191: The squares, in particular, are public spaces characterised by (1) connectivity, they are nodes of a network of streets that permits the movement through the urban environment;(2) aggregation, they are not just places to cross, but also places to stay, with a fundamental socio-cultural and economic role, hosting formal and informal community events along with services and economic activities; (3) healthiness, they have to be comfortable places. These kinds of structured lists make the reader think why these characteristics are particularly emphasizes, so references are needed.
  • Lines 197-203. I would consider to moving this description of MCDA tools into Introduction section. Methods section should be straight-forward description of the methods used in this particular study.
  • Since the main result of the paper is the approach and tool created by the authors, I would suggest that the sections 2.2.2 Mapping, 2.2.3 Evaluation, 2.2.4 Elaboration and 2.2.5 Scenario would already be under another title “Indicator-creation process” or related as section 3 (3.1. Mapping, 3.2. Evaluation…). That would leave the study design with the Figure 3 in the Methods section, and more content in the “Results” section. In these kinds of methodology-oriented papers the methodology is the actual result but it does not necessarily have to be titled as “Results”.
  • Lines 231-237. Consider discarding this text part or moving to Introduction section in a more compact form.
  • Figure 4. Please add a north arrow and scale bar.
  • Line 313. The tables contain… à which tables? Please refer to the exact tables.
  • Line 345: In the second evaluation phase… à In the evaluation phase? (there was only one, right?)
  • Figure 5 c and d would benefit from a legend.
  • Lines 348-355. Please explain in detail how and based on what the scoring was done. Was there a range for giving points, or was it made by point allocation of a certain number of points etc.?
  • A very important information is missing: Who did the scoring? The research group? How do you find that compatible from this research’s aims’ perspective that it was the researchers who both selected and valued the different indicators? Shouldn’t that be made by the stakeholders/planners?
  • Lines 389-390: Finally, to achieve a synthetic evaluation, the indicators scores have been integrated in a rating related to the public domain through a system of weights. The meaning of this sentence is very unclear, please revise or elaborate.
  • Lines 146: …the task for the students was to formulate… Who were these students and why did they participate this stage of the research?
  • Figure 10: Why is only the high school in the north-eastern corner of the park coloured blue? Please highlight the square also in this map. Please name the objects to the map, or provide a legend.

  • Results

  • This section is about your plans of conducting further research. This does not belong to this paper at this extent, and certainly is not results of this paper. Please provide a compact explanation of the future stakeholder engagement in the end of the previous section and delete this whole Results section.

4 Discussion

  • This section completely lacks discussion and reflection of your method with other scholars’ research in the same topic, other indicator sets etc. Please rewrite this whole section from this point of view. Discussion needs to focus on the creation process of the indicators, not engagement of stakeholders or practitioners, because that has not been done in this study.

Author Response

Dear reviewer,

thank you for the review and the valuable advice and corrections.

Here it follows the detail of our review:

  • The authors need to refocus the aim and results of this paper towards the actual results presented in this study (indicator creation) and not stakeholder/planning system engagement. Consequently, the structure of the paper concerning Methods and Results needs some fine tuning. The scoring / weighting stage of the method needs more elaboration, and the role of stakeholders (or its absence) needs to be clarified.
    • After a careful consideration of the suggested change in the paper’s structure, we have decided to keep the focus on the workflow as a dialogic process, which include the indicator creation as a step to engage the stakeholders. We do not think that the SI is a result per se: to do so, it would have required a more structured formalization. On the contrary, at this step, the SI remains intentionally flexible, coherently with the idea to discuss and integrate it in a participatory process. As stated in the results (l. 536-543), the stakeholders will be directly involved in the next phase of the project, and the workflow presented in the paper has permitted to set a starting point and define the next steps.
  • The weakest part of the paper is the Discussion section that completely lacks discussion with other scholars’ work;
    • The discussion section has been reviewed, maintaining the intention expressed in 2.1.1;
  • In addition, a further language and grammar check is still needed.
    • See point 1.1.1;
  • Lines 45-46: Consequently, their aesthetic/environmental quality has a strong impact on the general healthiness of people. Some scientific references are needed to mention to support this claim.
    • We have added the requested references;
  • Line 66. What is the paradigm shift that is referred to here? Is it the one described in the next paragraphs? In that case, this paragraph should maybe merged with the next.
    • We have changed to the plural ‘shifts’, and, in the following paragraphs, we touch the most relevant turning points for our topic;
  • Lines 67 and 69: The words Century, Mathematics, Economy, Geography, and Social Sciences should not be capitalized
    • the revision has been accepted;
  • Line 75: XX century. Previously used the notation 20th century. Choose one notation and use it coherently.
    • See 1.2.1;
  • Lines 95-99: (…) These three claims certainly need some references from the academic literature.
    • We have added the requested references;
  • Line 122: local PA. Please provide an explanation for the acronym PA
    • We added the explanation for the acronym the first time it appears;
  • Line 122-123. the proposed SIs hardly match with the procedures of decision-makers. This claim needs a reference.
    • We have added the requested references;
  • Since the paper concerns a spatial MCA tool, some recent scientific literature on those tools should be introduced in the Introduction section.
    • We have added the requested references;
  • The Introduction section is quite long, so maybe the description of the historical paradigm shifts could be made more compact in addition.
    • After a careful consideration of the suggestion, we have decided to not reduce the Introduction: it gives the opportunity to create a multidisciplinary theoretical and methodological framework and it does not seem unbalanced, in comparison to the other sections;
  • Figure 1: Maps a, b, and c lack a scale bar. The map d would benefit from a legend. The colour for the green spaces could be brighter for them to stand out from the base map. The isochrones visualisation of the 15-min city does not currently work, and it seems a bit unnecessary element in the map. Since the manuscript includes quite a lot of figures and maps, I would strongly suggest the authors to combine the maps Figure1 d and Figure 2.
    • We have removed the isochrones from map d. The maps a, b and c are more localization diagrams than proper cartographies and we think that in these cases a scale bar would disturb the readability rather than increasing it. The map d includes just a few layers (roads, buildings, water, parkings, green areas), which, in our opinion, do not require an explanation in a specific legend. After a careful consideration of the suggestion to merge map 1d and 2, we decided to keep them separate, to maintain a distinction between the presentation of the city as it is and the beginning of the definition of the project areas.
  • Line 184: The adopted methodological approach for answering this task is framed.... Please elaborate what task. You should not refer to other sections in the sentences in another sections.
    • Modified according to the suggestion;
  • Lines 186-191: (…) These kinds of structured lists make the reader think why these characteristics are particularly emphasizes, so references are needed.
    • In this case, the characteristics are selected by the authors according to the references related to public spaces proposed in the introduction.
  • Lines 197-203. I would consider to moving this description of MCDA tools into Introduction section. Methods section should be straight-forward description of the methods used in this particular study.
    • After a careful consideration of the suggestion, we decide to keep the paragraph in the previous position: it is more a methodological reference for the task proposed by the PA than a theoretical background of the research;
  • Since the main result of the paper is the approach and tool created by the authors, (…)
    • See 2.1.1
  • Lines 231-237. Consider discarding this text part or moving to Introduction section in a more compact form.
    • We have considered the suggestion, but we think that these lines work as an introduction to the mapping section: in the introduction there still are some references to this topic (l. 101-147).
  • Figure 4. Please add a north arrow and scale bar.
    • Done
  • Line 313. The tables contain… à which tables? Please refer to the exact tables.
    • Done
  • Line 345: In the second evaluation phase… à In the evaluation phase? (there was only one, right?)
    • We accepted the revision;
  • Figure 5 c and d would benefit from a legend.
    • We added the requested legend
  • Lines 348-355. Please explain in detail how and based on what the scoring was done. Was there a range for giving points, or was it made by point allocation of a certain number of points etc.?
    • Done (l. 357);
  • A very important information is missing: Who did the scoring? The research group? How do you find that compatible from this research’s aims’ perspective that it was the researchers who both selected and valued the different indicators? Shouldn’t that be made by the stakeholders/planners?
    • As stated in l. 350-352, the scoring has been performed by the research team. As stated in l. 536-543, the first expert base SI has the aim to activate the dialogue with the stakeholders, that will participate to the introduction of new indicators or to the discussion of the weight of the existing ones.
  • Lines 389-390: Finally, to achieve a synthetic evaluation, the indicators scores have been integrated in a rating related to the public domain through a system of weights. The meaning of this sentence is very unclear, please revise or elaborate.
    • We rephrased the sentence;
  • Figure 10: Why is only the high school in the north-eastern corner of the park coloured blue? Please highlight the square also in this map. Please name the objects to the map, or provide a legend.
    • The figure shows the technical map provided by the Municipality, with the highlight of the public services/buildings. The aim is just to show the structure of the square and we prefer not to add other labels/information to maintain the readability of a technical map.
  • This section is about your plans of conducting further research. This does not belong to this paper at this extent, and certainly is not results of this paper. Please provide a compact explanation of the future stakeholder engagement in the end of the previous section and delete this whole Results section.
    • In addition to the decision explained in 2.1.1, we do not understand where this section is about further research: apart of l. 536-543, where we contextualize the results in the general framework of an on going research, from l. 544 till the end of the section, we synthetize and present the outcomes of the proposed workflow;
  • Discussion: This section completely lacks discussion and reflection of your method with other scholars’ research in the same topic, other indicator sets etc. Please rewrite this whole section from this point of view. Discussion needs to focus on the creation process of the indicators, not engagement of stakeholders or practitioners, because that has not been done in this study.
    • We added some references to better contextualize the outcomes of our paper in a wider debate. Coherently with 2.1.1 explanation, and with the idea of the Discussion section as the space to present the next steps of the research, we keep the focus on the further expected development.

Reviewer 3 Report

This work is very interesting in its purpose and methodological approach. The authors present very clearly the objectives of the study, as well as the workflow of the investigation process. Discussion and results are consistent with the proposed objectives. The authors' methodological approach to the requalification of public open spaces in urban environments is very interesting and original in my opinion, which can be a valuable tool for decision makers in planning and land use.

I consider the document very well written and organized.

Just small issues that can be fixed:

  1. When the figures are originals, it does not require mentioning [Source: Authors].
  2. Perhaps in Table 1 the formats should be specified in the legend (for example SHP, shapefile format or GeoJSON is an open standard format designed for representing simple geographical features; It seems relevant to include some metadata in this table, in particular the level of spatial resolution or scale of each layer of information;
  3. line 251: instead of "univoval geographic projection" replace with "common spatial reference system"
  4. line 309 replace “geographic projection” with “cartographic projection”
  5. Table 2, replace column name “table” with “layer”
  6. Figure 10 – Replace “Technical map” with “Topographic map”

Two questions that could be used in future work by the team:

  1. data from social networks such as Twitter (e.g. the number of georeferenced tweets emitted through smartphone devices in a given location) to study the spatio-temporal dynamics of the population in an urban environment and infer about preferred urban places for people to stay? Could this be an important data source?
  2. I didn't see mentioned in this study any indicator related to security . Security in the sense of considering an open space a suitable place for evacuation in an emergency or catastrophe, or, conversely, considering open spaces as places prone to a higher incidence of crime (for examplo thefts). In fact, this last aspect is related to the feeling of security or insecurity that a place can transmit to people. Is this not also an indicator to be taken into account in addition to those considered by the authors?

Just reflection questions! Nothing that should alter or modify the approach followed by the authors.

Congratulations for this interesting work.

Author Response

Dear reviewer,

thank you for the review and the valuable advice and corrections.

Here it follows the detail of our review:

  • When the figures are originals, it does not require mentioning [Source: Authors].
    • We evaluated the suggestion, but we prefer to explicit the source for each figure;
  • line 251: instead of "univoval geographic projection" replace with "common spatial reference system"
    • Revision accepted;
  • line 309 replace “geographic projection” with “cartographic projection”
    • Revision accepted;
  • Table 2, replace column name “table” with “layer”
    • Revision accepted;
  • Figure 10 – Replace “Technical map” with “Topographic map”
    • The source is the technical map provided by the Municipality;
  • Perhaps in Table 1 the formats should be specified in the legend (for example SHP, shapefile format or GeoJSON is an open standard format designed for representing simple geographical features; It seems relevant to include some metadata in this table, in particular the level of spatial resolution or scale of each layer of information;
    • We added the requested information in note.

Round 2

Reviewer 2 Report

I am glad to see the authors have worked on improving the manuscript. Some suggestions were accepted, and some were rejected with good explanations why. However, I see that drastic improvements still need to be done for this publication to be scientifically valid. Especially the Results and Discussion sections need to be revised.

Major remarks and comments:

Focus of the manuscript

  • The focus of this particular manuscript cannot be the whole proposed process including participation of stakeholders, since that has not been done yet (even though it will be done in at a later stage of their project). The authors need to focus their manuscript on the results (the methodology) that has been created and presented in this paper; what has been done so far (indicator creation), not what is going to be done next. That is why the authors need to significantly reduce the text parts that imply the future participation of stakeholder and their knowledge. For example, the claim in the Introduction section (lines 128-131) needs to be revised:

“The core of our proposal is to consider the SI not a prescriptive but a dialogic tool; a base for collecting and integrating input from the different involved actors in a participatory process that includes researchers, experts, stakeholders, and citizens.”

-->

“The core of our proposal is to consider the SI not a prescriptive but a dialogic tool; creating a base for collecting and integrating input from the different involved actors in a participatory process at a later stage of its implementation.” (or related)

  • The research group did the indicator scoring, which does not make this method very participatory at this point (the scoring could have already been participatory), but if I understand correctly, the scores will be adjusted with stakeholders in the future. The fact that scores will be adjusted with stakeholders at later stage of the method could be clearly stated in the method/results section because it increases the credibility of this study.

Results section

  • I am aware that structuring this kind of methodology-oriented paper is difficult, and I understand the decision of the authors not to include the methodology sections under the Results section. However, the beginning of the Result section still misleads the reader. It should start with your current results presented in this paper, not future development of the method, so please move this text part discussing your future plans in the end of the chapter:
  • “The proposed workflow has provided dialogic inputs for the next participatory phase that will involve the research team and the PA. The SI sets the base for integrating new parameters, according to the instances of the involved stakeholders, adjusting, consequently, the evaluative synthesis and the decision priorities.”
  • And start the section like this (or related): The results of the synthetic evaluation provided by the expert-based SI presented in this article, has revealed three main recurrent patterns, identifying three typologies of squares and, consequently, allowing developing ad hoc design strategies…

Discussion section

  • The authors have added some references to Discussion section, but in general; it still needs to be improved. The interests of academic audience and topics that you should discuss are for example: comparing your results on the development needs of squares with other scholars who have studied them: are there similar kinds of planning problems / solutions detected? How your method differs from other similar approaches (MCDA) or indicator sets or does it? The discussion section in a scientific article cannot lean only about the future development of the method that is presented in the current paper; but to reflect your results with other studies.

 Minor comments and questions:

 2.1 The case study

  • I would still strongly encourage the authors to merge Figures 1d and 2. It would actually make the article more readable for readers if there was only one map showing the location of both the study area and squares.

2.2.2 Mapping

  • In the Figure 5: Please note that unit symbol of metre is m, not m

Language

  • My previous comment: a further language and grammar check is still needed.
  • Author response: ”See point 1.1.1;”
  • What is point 1.1.1?

Author Response

Dear reviewer 2,

thanking the reviewer for the precious contribution, and after a careful review of the paper, we add some notes about the major remarks and critical points to better explain our position.

  • Focus of the manuscript: as stated in the abstract, «the paper presents an overview of the workflow, with a focus on the first set of thematic indicators and an open conclusion» (l. 21-22). The presentation of the whole workflow, with the specification of the current state and the next phases, is fundamental to understand that the first set of thematic indicators is not a conclusion per se, but it is an activation step of a wider process, that has started, in dialogue with the public administration, with the input domains, and will continue with the direct involvement of different stakeholders.

In fact, «the core of our proposal is to consider the SI not a prescriptive but a dialogic tool; a base for collecting and integrating input from different actors in a participatory process» (l. 129-130). We accept to rephrase the sentence, maintaining the idea that the SI is functional to the next direct involvement of the stakeholders and it cannot be abstracted from the workflow.

We did the first SI and the scoring to provide the stakeholders with a sample, a flexible input to be elaborated, adjusted and integrated, as stated in the introduction of the results: «the SI sets the base for integrating new parameters, according to the instances of the involved stakeholders, adjusting, consequently, the evaluative synthesis and the decision priorities» (l. 575-577).

  • Results: we accept the reordering suggestion.
  • Discussion: we integrate the section with further references to other similar works/approaches.

Best regards,

This manuscript is a resubmission of an earlier submission. The following is a list of the peer review reports and author responses from that submission.